# A conserved enzyme of smut fungi facilitates cell-to-cell extension in the plant bundle sheath

Bilal Ökmen [1,2] ✉, Elaine Jaeger[1], Lena Schilling[3], Natalie Finke[1], Amy Klemd [3], Yoon Joo Lee [1], Raphael Wemhöner[1], Markus Pauly [4], Ulla Neumann[5] & Gunther Doehlemann [1] ✉

Smut fungi comprise one of the largest groups of fungal plant pathogens causing disease in all cereal crops. They directly penetrate host tissues and establish a biotrophic interaction. To do so, smut fungi secrete a wide range of effector proteins, which suppress plant immunity and modulate cellular functions as well as development of the host, thereby determining the pathogen's lifestyle and virulence potential. The conserved effector Erc1 (enzyme required for cell-to-cell extension) contributes to virulence of the corn smut *Ustilago maydis* in maize leaves but not on the tassel. Erc1 binds to host cell wall components and displays 1,3-β-glucanase activity, which is required to attenuate β-glucan-induced defense responses. Here we show that Erc1 has a cell type-specific virulence function, being necessary for fungal cell-to-cell extension in the plant bundle sheath and this function is fully conserved in the Erc1 orthologue of the barley pathogen *Ustilago hordei*.

Plants have established physical and chemical defense layers to protect themselves from continuous pathogenic attacks. The plant cell wall is one of the main physical barriers protecting plants against such attacks. In turn, phytopathogenic microorganisms developed strategies to breach the host cell wall allowing successful host penetration and colonization. In this regard, some filamentous phytopathogens use both mechanical force and enzymatic activities to breach this first layer of defense. While a specialized dome-like structure, the appressorium, provides mechanical force for direct penetration, each pathogen has an arsenal of plant cell wall degrading enzymes (PCWDEs) to break down the host cell wall for host penetration or nutrient acquisition[1]. Plant pathogens show a wide range of variation in their PCWDEs repertoire including glycoside hydrolases (GHs), which determines their virulence, pathogenic lifestyle, and host specificity[2–4]. Compared to necrotrophic and hemibiotrophic pathogens, biotrophs possess relatively few PCWDEs, which is in line with their need to preserve the cell wall integrity of their host plant cells and minimize

release of damage-associated molecular patterns (DAMPs)[5]. DAMPs are endogenous molecules, including plant cell wall components and peptides, that are released from a host plant upon damage or infection. They serve as signaling molecules to induce defense-related responses against invading pathogens and promote damage repair[6].

*Ustilago maydis*, the causative agent of corn smut disease, is a biotrophic fungal pathogen that induces tumor formation in all aerial plant organs, including leaves, ears, and tassels. Like other phytopathogenic smut fungi, *U. maydis* grows both extra- and intracellularly and extends from cell-to-cell to colonize its host[7]. To establish disease, *U. maydis* secretes effector proteins, which are deployed in an organ-specific manner[8,9]. Differences in the arrangement of tissue types, cell types, in cell wall components or in metabolite compositions may necessitate organ-specific effectors[10].

Schilling et al. described a set of leaf-specific *U. maydis* effectors of which one (UMAG_01829) is predicted to be an α-L-arabinofuranosidase and shows the highest expression level among

[1]Institute for Plant Sciences, University of Cologne, BioCenter, Zuelpicher Str. 47a, 50674 Cologne, Germany. [2]Department of Microbial Interactions, IMIT/ZMBP, University of Tübingen, Tübingen, Germany. [3]Max-Planck-Institute for Terrestrial Microbiology, 35043 Marburg, Germany. [4]Institute for Plant Cell Biology and Biotechnology, Heinrich Heine University of Düsseldorf, Universitätsstr. 1, D-40225 Düsseldorf, Germany. [5]Central Microscopy, Max-Planck-Institute for Plant Breeding Research, 50829 Cologne, Germany. ✉e-mail: bilal.oekmen@zmbp.uni-tuebingen.de; g.doehlemann@uni-koeln.de

the *U. maydis* effector genes[8,11]. The presence of a putative PCWDE among the organ-specific effectors might reflect differences in cell wall composition between leaf and tassel tissues. Several studies have demonstrated the importance of pathogen-derived PCWDEs in plant–microbe interactions[1]. While some glycoside hydrolases are involved in host penetration and nutrient acquisition by degrading plant cell wall components, others are involved in the detoxification of antimicrobial secondary metabolites or the hydrolysis of microbe-associated molecular patterns (MAMPs)[1]. For example, xylanases have been shown to be involved in the proliferation of *U. maydis* during plant infection[12]. While a tomatinase enzyme of the GH10 family from *Cladosporium fulvum* is required for detoxification of tomato-specific α-tomatine[13], a chitinase of the GH18 family from *Magnaporthe oryzae* (MoChia1) binds and sequesters chitin fragments that are released from the fungal cell wall to prevent MAMP-triggered immunity[14]. Likewise, several arabinofuranosidases have been described to contribute to the virulence of plant pathogens. For example, the endo-arabinase BcAra1 is necessary for the full virulence of the necrotrophic plant pathogen *Botrytis cinerea* in *Arabidopsis thaliana*[15]. In *Sclerotinia trifoliorum*, arabinofuranosidase-deletion mutants display reduced virulence in *Pisum sativum var. avense*[16]. α-L-arabinofuranosidases are enzymes that catalyze the hydrolysis of terminal, non-reducing α-L-arabinofuranose residues in α-L-arabinosides. The enzyme works on terminal α-L-1,2-; α-L-1,3- and α-L-1,5-arabinofuranosyl residues of cell wall polymers, such as arabinoxylans and arabinogalactans[17]. Arabinofuranosidases often act in concert with other hemicellulases to degrade the hemicellulose of the plant cell wall[18].

In this work, we have functionally characterized Erc1 (enzyme required for cell-to-cell extension), a conserved effector of smut fungi with an organ-specific virulence function. Despite a predicted α-L-arabinofuranosidase activity based on protein sequence similarity, we found that Erc1 exhibits 1,3-β-glucanase activity and is required for cell-to-cell extension specifically in bundle sheaths cells. Strikingly, this cell-type specific virulence function is conserved in the Erc1 orthologue of the barely pathogen *Ustilago hordei*. This identifies Erc1 as a secreted enzyme with a highly specific virulence function, which is conserved amongst different pathosystems.

## Results

### UMAG_01829 is a leaf-specific virulence factor of *Ustilago maydis*

In a previous study[8], we have shown organ-specific virulence activities of predicted effector protein from *U. maydis*. Among the effector genes being specifically required for tumorigenesis in maize leaves, *UMAG_01829* caught our attention because of its predicted protein properties and function, which are rather atypical for a fungal effector: *UMAG_01829* encodes a 703 amino acid-long protein with an N-terminal secretion signal (1-18 aa; SignalP 5.0), followed by a predicted carbohydrate-binding module (CBM, aa 124-266) (Supplementary Fig. 1a). The C-terminal part of the protein (aa 272-691) is predicted to carry a catalytic domain of the GH51 Family (α-L-arabinofuranosidase) with two predicted active sites at amino acids 410-412[GNE] and 499[E] (Supplementary Fig. 1a). Hydrolases of the GH51 family have been shown to exhibit mainly hemicellulose, α-L-arabinofuranosidase activity, i.e., they remove terminal arabinosyl-moieties from arabinoxylans or other branched arabinans, but they can also display endoglucanase and endoxylanase activities (cazy.org). The *U. maydis* genome has additional homologs of α-L-arabinofuranosidase (*Afg*)-encoding genes, which belong to GH51 (Afg2; UMAG_00837) and GH62 (Afg3; UMAG_04309, a commercially available α-L-arabinofuranosidase). Neither Afg2 nor Afg3 contains a predicted CBM (Supplementary Fig. 1b) and they do not contribute to *U. maydis* virulence on maize (Supplementary Fig. 2b)[19]. All three hydrolase genes are strictly expressed only *in planta* (Supplementary Fig. 2a). While *UMAG_01829* is one of the most highly expressed effector genes in *U. maydis* at all

tested time points of colonization, *Afg2* is lowly expressed only at the very early and very late stage of infection (12 days post infection (dpi))[11]. Like *UMAG_01829*, *Afg3* is expressed throughout the leaf infection process; however, the expression level of *UMAG_01829* is significantly higher when compared to *Afg3* (Supplementary Fig. 2a)[11]. Homology search and phylogenic tree analysis showed that Afg2 and Afg3 are widely conserved in fungal and bacterial genomes. In contrast, UMAG_01829 is highly conserved within Ustilaginomycotina, but it is not found outside the Basidiomycetes (Supplementary Fig. 3).

### UMAG_01829 is required for fungal cell-to-cell extension in the host bundle sheath

As a first step of functional characterization, we confirmed the previous finding of ref. 8, showing that UMAG_01829 is required for full virulence of *U. maydis* in maize leaves (Fig. 1a). Genetic complementation of the *U. maydis* SG200ΔUMAG_01829 mutant with the native *UMAG_01829* gene fully restored *U. maydis* virulence, confirming that the observed defect in virulence was solely caused by the deletion of *UMAG_01829*. To investigate which step of host infection is affected by *UMAG_01829* deletion, fungal growth inside the leaf was followed by confocal microscopy. SG200ΔUMAG01829 did not show any defect in early pathogenic development. Neither appressoria formation, nor epidermal penetration was impaired when compared to the SG200 progenitor strain (Supplementary Fig. 2c, d). Contrary, the SG200ΔUMAG_01829 mutant displayed a defect in cell-to-cell extension and this phenotype specifically appeared in bundle sheath cells (Fig. 1b, c). While 67% of cell-to-cell extension attempts of the SG200ΔUMAG_01829 mutant failed in bundle sheaths cells, this was only observed in 17% of the attempts of both SG200 and SG200Δerc1/C complementation strains (Fig. 1b). Because of this specific function in cell-to-cell extension, we named UMAG_01829 UmErc1 (*Ustilago maydis* **e**nzyme **r**equired for **c**ell-to-cell extension).

To test whether the predicted carbohydrate-binding module (CBM) and its putative catalytic activity are required for its virulence function, various mutant forms of Erc1 including one without the CBM domain, as well as an active site mutant (Erc1[M2x]: 410-412[GNE>AAA] and 499[E>A]) were expressed in the *U. maydis* Δerc1 mutant background. To test for virulence complementation, maize seedlings were inoculated with *U. maydis* SG200, SG200Δerc1 mutant (Δerc1), SG200Δerc1/Erc1 (Δerc1/C), SG200Δerc1/Erc1[ΔCBM] (ΔCBM) and SG200Δerc1/Erc1[M2x] (Δerc1/C[M2x]) strains (Figs. 1a and 2a). While expression of the wild-type Erc1 fully recovered the virulence phenotype of SG200Δerc1 mutant at 12 dpi, expression of either Erc1[M2x] or Erc1[ΔCBM] only partially rescued the reduced virulence phenotype of the SG200Δerc1 mutant at 12 dpi (Figs. 1a and 2a). Site-directed mutagenesis at the two predicted active sites of the catalytic domain (Erc1[M2x]: 410-412[GNE>AAA] and 499[E>A]) resulted in a significant reduction in virulence compared to the progenitor strain. However, there was no significant difference between the SG200Δerc1 and SG200Δerc1/Erc1[M2x] strains (Fig. 1a), indicating that enzymatic activity of Erc1 is required for its virulence function.

### Erc1 is functionally conserved in covered smut of barley

Homology search and phylogenetic tree analysis revealed the presence of *Erc1* homologs in all available genomes of smut fungi (Supplementary Figs. 1 and 3). Considering the high degree of sequence conservation, we asked whether this effector could be functionally conserved in smut fungi, despite its highly specific virulence function in *U. maydis*. Thus, we expressed the *Erc1* gene from *Ustilago hordei*, the covered smut of barley, in the SG200Δerc1 mutant under the control of the native *U. maydis* Erc1 promoter. Disease assays performed with the *U. maydis* SG200, SG200Δerc1 mutant, SG200Δerc1/Erc1, and SG200Δerc1/UhErc1 strains showed that UhErc1 fully restored virulence of the SG200Δerc1 mutant (Fig. 2b). In parallel, we deleted the *UhErc1* gene in *U. hordei* to test its contribution to fungal virulence during barley colonization. Following inoculation of barley

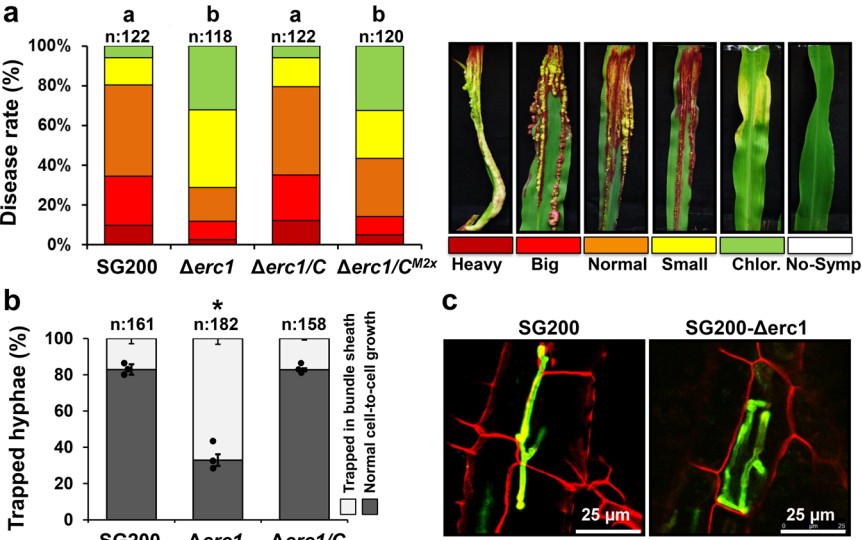

**Fig. 1 | Erc1 is a virulence factor that is involved in cell-to-cell extension in maize. a** Disease symptoms caused by *Ustilago maydis* SG200, SG200Δerc1 mutant, SG200Δerc1/C strain, and SG200Δerc1/Erc1[M2x] on Early Golden Bantam (EGB) maize leaves at 12 days post inoculation (dpi). Disease rates are given as a percentage of the total number of infected plants. n: indicates total number of infected maize seedlings in three independent biological experiments and letters above bars indicate significant differences (One-way ANOVA followed by Tukey multiple comparison test was performed, *p* < 0.05). **b** Quantification of cell-to-cell penetration efficiency of SG200, SG200Δerc1 and SG200Δerc1/C complementation strains. The graph depicts the percentage of trapped *U. maydis* hyphae in maize bundle sheath cells at 4 dpi. n: indicates total number of counted infected maize cells in three independent biological experiments. Asterisks above bars indicate significant differences (*p* < 0.05, Chi-square test). Data are presented as mean value ± SD. **c** Microscopic observation of trapped *U. maydis* SG200 and SG200Δerc1 hyphae in maize bundle sheath cells at 4 dpi via WGA-AF488/Propidium iodide staining. WGA-AF488 (green color -fungal cell wall): excitation at 488 nm and detection at 500–540 nm. PI (red color - plant cell wall): excitation at 561 nm and detection at 580–630 nm. Similar results were observed at least in three independent biological experiments. Calculated *p* values are shown in the Source Data.

seedlings with *U. hordei* strains DS199, DS199Δerc1, and DS199Δerc1/Erc1, we observed a significantly reduced fungal biomass of DS199Δerc1 compared to DS199 and the DS199Δerc1/Erc1 complementation strains (Fig. 2c, d). Moreover, the DS199Δerc1 mutant showed a cell-arrest phenotype in the bundle sheath cells in barley leaves, mirroring the phenotype of the corresponding *U. maydis* mutant (Fig. 2e, f). 80% of cell-to-cell extension attempts of the DS199Δerc1 mutant failed in bundle sheath cells compared to 20% for both DS199 and DS199Δerc1/Erc1 complementation strains (Fig. 2e, f). Together, these results demonstrate that the cell-type specific virulence function of Erc1 is conserved among the maize pathogen *U. maydis* and the barley pathogen *U. hordei*.

### Erc1 is secreted to the biotrophic interface

To localize Erc1 during host colonization, Erc1 with a fused C-terminal mCherry tag was expressed in the SG200Δerc1 mutant under the control of its native promoter. Confocal microscopy was performed using maize leaves inoculated with the Erc1-mCherry at 2 dpi. The SG200 strain expressing Pit2-mCherry, an effector that was previously shown to localize in the maize apoplast[20], was used as a positive control for secretion. The SG200 strain Int.mCherry expressing cytosolic mCherry served as a negative control (Fig. 3a). Confocal microscopy confirmed that both Erc1-mCherry and Pit2-mCherry showed fluorescent signals on the outer surface of fungal hyphal tips, while the Int.mCherry showed fluorescent signals only inside the fungal cell (Fig. 3a). After plasmolysis with 1 M sodium chloride solution, the Erc1-mCherry signals accumulated in the apoplastic space and at the site of cell-to-cell passage on the plant cell wall, indicating extracellular secretion of Erc1-mCherry (Fig. 3a). For a more detailed subcellular localization of Erc1, we performed transmission electron microscopy (TEM) of immunogold labeled maize leaf sections infected by an Erc1-HA expressing SG200 strain (Fig. 3b). The SG200 strain expressing secreted GFP-HA was used as a control to compare signal distribution and specificity. TEM micrographs obtained from immunogold labeling

with a monoclonal antibody for HA depicted secretion of Erc1-HA to the biotrophic interface. Erc1-HA and GFP-HA signals were detected both in the fungal cell wall and in adjacent plant cell wall regions (Fig. 3b). No significant differences in signal distribution between Erc1-HA and GFP-HA could be observed, which confirmed secretion and apoplastic localization of Erc1 and ruled out any specific focal accumulation of the effector.

### Erc1 does not exhibit α-L-arabinofuranosidase activity

To biochemically characterize the *U. maydis* Erc1 protein, N-terminally His tagged and C-terminally Myc/His-double-tagged Erc1 and its predicted active site mutants (Erc1[M2x]: 410-412[GNE>AAA] and 499[E>A]) were produced in the *Pichia pastoris* protein expression system. Subsequently, the recombinant proteins were purified via Nickel-NTA affinity chromatography (Fig. 4a). On SDS-PAGE, Erc1 recombinant protein appeared as two bands with higher molecular weight than the expected suggesting post-translational modifications (Fig. 4a). A western blot analysis of the recombinant His-Erc1-Myc-His protein with antibodies specific for the His-tag and Myc-tag showed that, while the upper protein band was detectable both with an α-His and α-Myc antibody, the lower protein band was detectable only with an α-His antibody indicating a C-terminal cleavage of the tags (Fig. 4a).

Moreover, although both lower and upper Erc1 bands occurred in similar protein amounts in a Sypro Ruby stained gel, the signal intensity of the upper band with the α-His antibody was much higher than the lower band, indicating also partial N-terminal cleavage (Fig. 4a).

To test for the predicted α-L-arabinofuranosidase activity of Erc1, an enzyme activity assay was carried out with 4-nitrophenyl-α-L-arabinofuranose as a substrate. This substrate was incubated with either *U. maydis* Erc1 or a commercial arabinofuranosidase from *Aspergillus niger* (AFASE) as a positive control. While the commercial AFASE enzyme exhibited the expected α-L-arabinofuranosidase activity, no activity was detected in the samples incubated with the *U. maydis* Erc1 recombinant protein (Fig. 4b) suggesting that under the tested

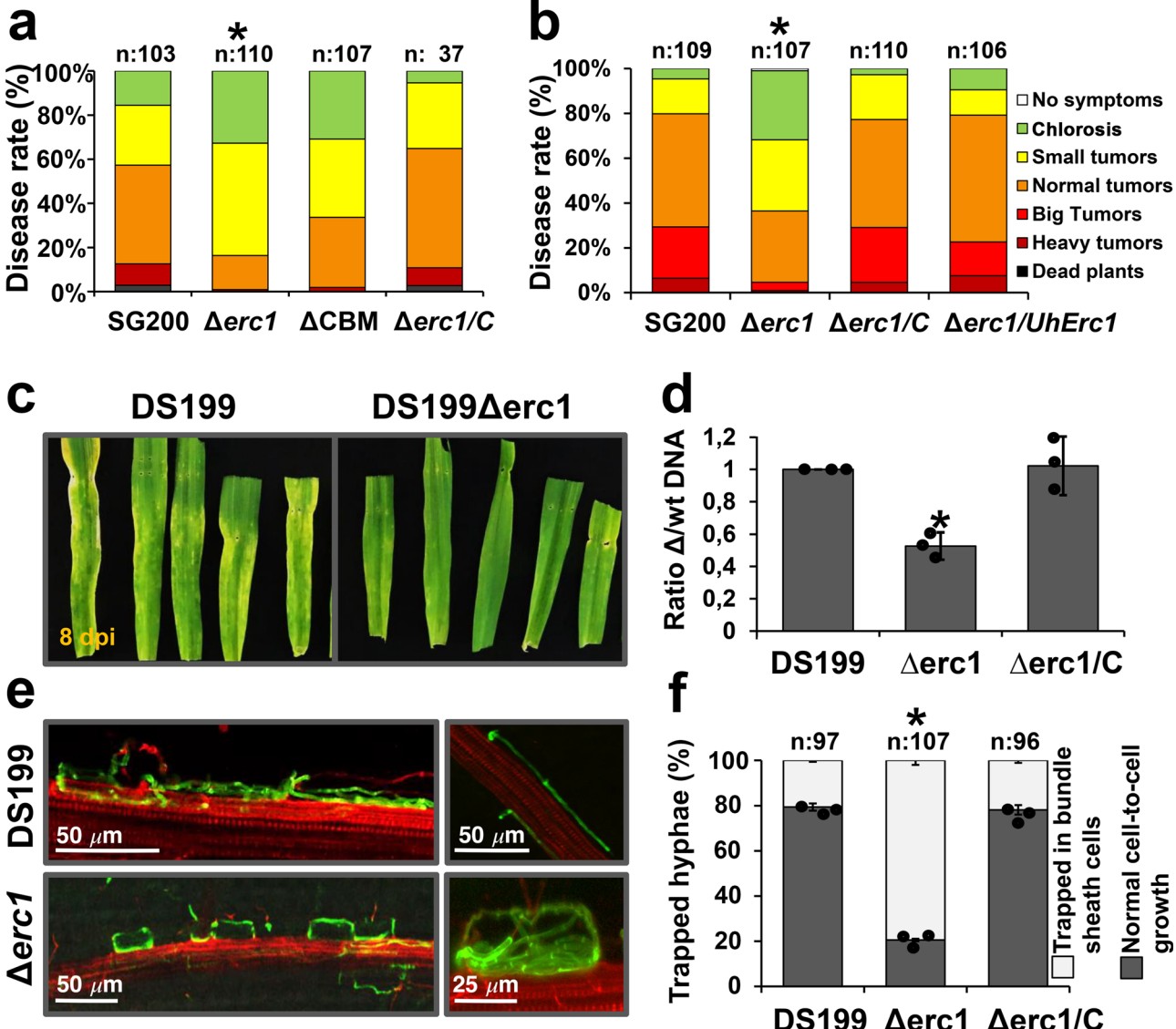

**Fig. 2 | Erc1 is functionally conserved in smut fungi. a** The carbohydrate-binding module (CBM) is required for full function of *Ustilago maydis* Erc1. Disease assay was performed for *U. maydis* SG200, SG200Δerc1, SG200Δerc1/C strain, and SG200Δerc1/Erc1ΔCBM on EGB maize at 12 days post inoculation (dpi). Disease rates are given as a percentage of the total number of infected plants. *n*: indicates total number of infected maize seedlings in three independent biological experiments. Asterisks above bars indicate significant differences (*p* < 0.05, two-tailed student's t-test). **b** Complementation of SG200Δerc1 mutant with *Ustilago hordei* UhErc1. Disease assay was performed for *U. maydis* SG200, SG200Δerc1, SG200Δerc1/C, and SG200Δerc1/UhErc1 strains on EGB maize plants at 12 days post inoculation (dpi). *n*: indicates total number of infected maize seedlings in three independent biological experiments. Asterisks above bars indicate significant differences (*p* < 0.05, two-tailed student's t-test). **c** UhErc1 is a virulence factor during barley infection. Disease assay was performed with *U. hordei* DS199 and DS199Δerc1 mutant strains on 13-day-old barley seedlings. Pictures were taken at 8 dpi. Similar results were observed in three independent biological experiments. **d** Quantification of fungal biomass of DS199, DS199Δerc1 mutant, and DS199Δerc1/C complementation strains on barley at 8 dpi. qPCR was performed to determine fungal biomass by using gDNA that was isolated from *U. hordei* DS199, DS199Δerc1 mutant, and DS199Δerc1/C infected barley leaves. Three independent biological replicates were performed with total number of 15 infected barley seedlings for each experiment. Data were presented as mean value ± SD. Asterisks above bars indicate significant differences (*p* < 0.05, two-tailed student's t-test). **e** Microscopic observation of trapped *U. hordei* DS199 and DS199Δerc1 mutant hyphae in barley bundle sheath cells at 8 dpi via WGA-AF488/Propidium iodide staining. WGA-AF488 (green color–fungal cell wall): excitation at 488 nm and detection at 500–540 nm. PI (red color–plant cell wall): excitation at 561 nm and detection at 580–630 nm. Similar results were observed in three independent biological experiments. **f** Quantification of cell-to-cell penetration efficiency of *U. hordei* DS199, DS1990Δuherc1 mutant and DS199Δerc1/C strains. The graph depicts the percentage of trapped *U. hordei* hyphae in barley bundle sheaths cells at 8 dpi. *n*: indicates total number of counted infected barley cells in three independent biological experiments. Data were presented as mean value ± SD. Asterisks above bars indicate significant differences (*p* < 0.05, Chi-square test). Calculated *p* values were presented in Source Data.

conditions and with this particular substrate Erc1 did not act as an α-L-arabinofuranosidase. In addition, an activity-based protein profiling (ABPP) assay was performed by using the α-L-arabinofuranosidase specific fluorescent probe (ME868) and two specific inhibitors (DL69 and DL85) as controls (Fig. 4c)[21]. In the ABPP assay, the fluorophore-tagged probe binds covalently and specifically to the active site of the enzyme. In this assay, the fluorescent signal level is positively correlated with the respective enzymatic activity, thus lower/no fluorescent signal in the presence of both hydrolase-specific probe and inhibitor indicates inhibition of a specific enzymatic activity. The ABPP assay showed that the ME868 probe specifically bound to the AFASE enzyme control, and pre-treatment of AFASE with α-L-arabinofuranosidase-

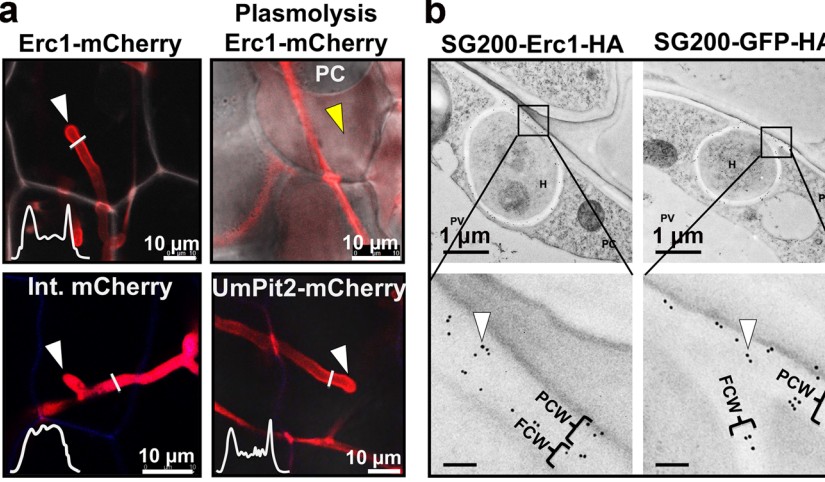

**Fig. 3 | Localization of Erc1 in *Ustilago maydis* SG200 during maize colonization. a** Erc1-mCherry was heterologously expressed in *U. maydis* SG200 strain under control of the native promotor with predicted native signal peptide for extracellular secretion. The SG200 strains expressing the Erc1-mCherry, UmPit2-mCherry (as a positive control for secretion) and cytosolic mCherry (int. mCherry; as a negative control for secretion) were inoculated on maize seedlings and at 4 dpi confocal microscopy was performed to monitor the localization of each recombinant protein. While both Erc1-mCherry and UmPit2-mCherry are secreted around the tip of the invasive hyphae, internal mCherry localizes to the fungal cytoplasm. The white graphs indicate the mCherry signal intensity along the diameter of the

hyphae (illustrated by white lines in the image). White arrowheads indicate fungal hyphal tips and yellow arrowhead indicates apoplastic fluid after plasmolysis. Plasmolysis was performed with 1 M NaCl solution. Similar results were observed in three independent biological experiments. **b** Transmission electron micrographs after immunogold labeling of secreted Erc1-HA and GFP-HA with a monoclonal antibody recognizing HA epitopes. White arrowheads pointing to black dots indicate secretion of Erc1-HA to the biotrophic interface. FCW: Fungal cell wall, H: Hyphae, P: Plant cell cytoplasm, PCW: Plant cell wall. Similar results were observed in multiple transmission electron micrographs.

specific DL69 and DL85 inhibitors prevented this binding demonstrating the specificity of the assay and confirming the α-L-arabinofuranosidase activity of AFASE (Fig. 4c). However, compared to AFASE only a very weak signal was observed for Erc1, which could not be inhibited by the α-L-arabinofuranosidase-specific inhibitors. Therefore, we concluded that the weak signal represented a non-specific background signal (Fig. 4c). In addition to the enzyme activity assays, we complemented the SG200Δerc1 mutant with *Afg2* and *Afg3* (encoding the commercially available α-L-arabinofuranosidase) being expressed under the control of the native *Erc1* promoter. The resulting strains were used to test whether an enzyme with α-L-arabinofuranosidase activity could rescue the virulence phenotype of the *Erc1* deletion mutant. However, disease assays with *U. maydis* strains SG200, Δerc1 mutant, Δerc1/C, Δerc1/Afg2, and Δerc1/Afg3 complementation strains showed that neither Afg2 nor Afg3 could complement the virulence defect of the Δerc1 mutant (Fig. 4d).

### Erc1 exhibits exo-β−1,3-glucanase activity

Since α-L-arabinofuranosidase activity of Erc1 could not be demonstrated and none of the two other *U. maydis* α-L-arabinofuranosidases could restore the Δ*erc1* mutant infection phenotype, we decided to test whether Erc1 might exhibit any other hydrolase activity that has been ascribed to some members of the GH51 family such as endoglucanase activity. To this end, we incubated the recombinant Erc1 protein with several polysaccharides including β−1,4-glucan, laminarin, lichenan, xylan, and arabinoxylan and used thin-layer chromatography (TLC) to visualize the released sugars from the tested candidate polymers. This approach revealed that Erc1 hydrolyzed laminarin, which consists of a linear β−1,3-glucan with β−1,6-branches. Thus, Erc1 exhibits β-glucanase activity (Supplementary Fig. 4a). Laminarin polysaccharides consist of 20-25 units of linear β−1,3-glucan with β−1,6-linkages. To test whether Erc1 cleaved the β−1,3-glucan backbone or the β−1,6-linked branches, we performed an assay using laminarihexaose, which contains only linear β−1,3-glucan linkages (Fig. 4e, f). The TLC results showed that Erc1 could hydrolyze laminarihexaose demonstrating a β−1,3-glucanase activity. Furthermore, TLC

performed with laminarin substrate incubated with Erc1, commercial exo-β-glucanase and endo-β-glucanase enzymes showed that, unlike the commercial endo-β-glucanase, which released several products with different intermediate sizes, both Erc1 and the exo-β-glucanase enzymes released only single glucose-moieties, suggesting an exo-β−1,3-glucanase activity of Erc1 (Supplementary Fig. 5a). To confirm the predicted active sites of Erc1, recombinant mutant variants of Erc1 carrying either single (Erc1M1x: 410-412GNE>AAA) or double (Erc1M2x: 410-412GNE>AAA and 499E>A) mutations in the active site were incubated with both substrates. TLC showed a reduced enzymatic activity compared to recombinant wild-type Erc1. The residual enzymatic activity of the mutated Erc1 suggested the presence of additional active site/s, which we could not identify in this study (Fig. 4e, f).

### Erc1 binds to plant cell wall components and suppresses laminarihexaose-induced ROS burst in plant leaves

To test the function of the predicted carbohydrate-binding module (CBM) of Erc1, we performed carbohydrate binding assays. Recombinant Erc1 protein was incubated with insoluble polysaccharides originating from either the fungal cell wall (chitin and chitosan) or the plant cell wall (cellulose, xylan, β−1,4-glucan, and lichenan). The subsequent pull-down assay showed that Erc1 did not bind any of the tested fungal cell wall components (Supplementary Fig. 5b). In contrast, it bound the plant cell wall components cellulose and lichenan, suggesting that Erc1 primarily binds the plant cell wall (Supplementary Fig. 5b). Co-incubation of Erc1 with either cellulose or lichenan did not produce any hydrolyzed products indicating that both polymers do not represent enzymatic substrates of Erc1 (Supplementary Fig. 4a). Monosaccharide analysis of the matrix polymers that are present in maize leaves following treatment with mock, SG200, or SG200Δerc1 *U. maydis* strains at 9 dpi did show differences between mock and SG200 infected leaf tissues as the wall arabinose content was increased and the wall matrix glucose was decreased. However, no differences in plant wall composition were observed between SG200 and SG200Δerc1 inoculations (Supplementary Fig. 4b). These data indicate that Erc1 does not quantitatively hydrolyze polymers in the plant cell wall.

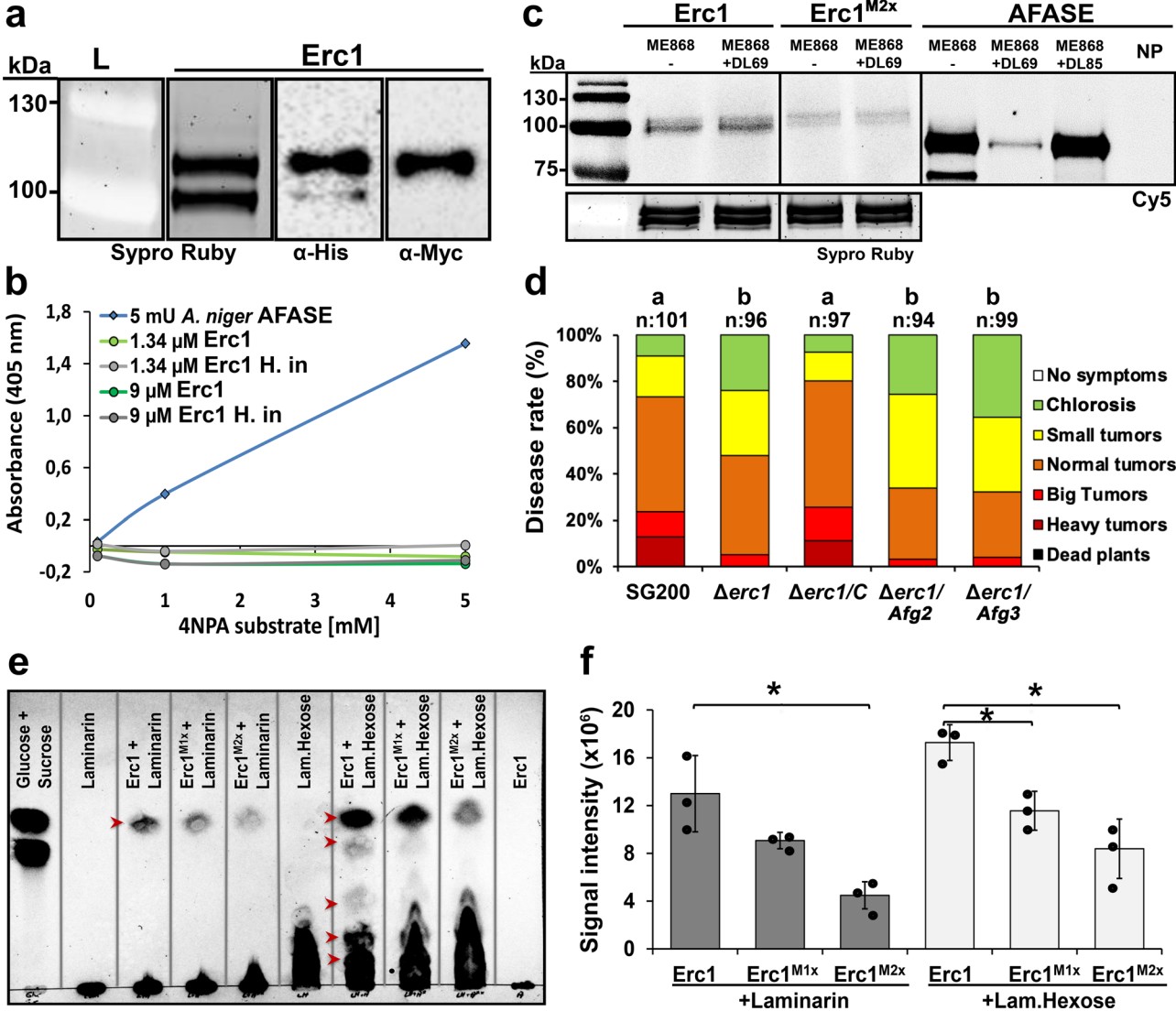

**Fig. 4 | Functional characterization of the Erc1 protein. a** Purification of His-Erc1-Myc-His recombinant protein. Sypro Ruby staining was performed to visualize the Erc1 recombinant protein. Western blot (WB) analysis was performed with α-His- and α-Myc- specific antibody to detect His-Erc1-Myc-His protein. While two bands were detectable in Sypro Ruby staining and WB performed with α-His, only one band was detectable with α-Myc-specific antibody. Similar results were observed at least in two independent biological replicates. **b** α-L-arabinofuranosidase activity assay with 4-nitrophenyl α-L-arabinofuranoside (4NPA) substrate. A commercial α-L-arabinofuranosidase from *Aspergillus niger* (AFASE) was used as a positive control. Data are presented as mean value of three independent biological experiments. **c** Activity-based protein profiling (ABPP) assay for Erc1. The Erc1, Erc1$^{M2x}$ and AFASE recombinant proteins were incubated with the specific α-L-arabinofuranosidase inhibitor DL69. Plus (+) and minus (−) indicate the addition and the absence of the inhibitor, respectively. α-L-arabinofuranosidase specific probe ME868 was added as indicated (+/·). The probe was detected by scanning the in-gel fluorescence with Cy5 filter (Ex. 650 nm, Em. 670 nm). Protein of loaded samples were visualized via Sypro Ruby (Ex. 450 nm, Em. 610 nm). NP: no probe control. Similar results were observed in two independent biological experiments. **d** Complementation of SG200Δerc1 mutant with *Afg2* and *Afg3* under the native

*Erc1* promoter. Disease assay was performed for *Ustilago maydis* SG200, SG200Δerc1 mutant, SG200Δerc1/C, SG200Δerc1/Afg2 and SG200Δerc1/Afg3 strains on EGB maize plants at 12 days post inoculation (dpi). Disease rates are given as a percentage of the total number of infected plants. n: indicates total number of infected maize seedlings in three independent biological replicates. Letters above bars indicate significant differences (One-way ANOVA followed by Tukey multiple comparison test was performed, $p < 0.05$). **e** Thin layer chromatography (TLC) assay was performed to demonstrate β−1,3-glucanase activity of Erc1 on laminarin and laminarihexaose substrate. Erc1, Erc1$^{M1x}$, and Erc1$^{M2x}$ recombinant proteins were incubated with laminarin and laminarihexaose. Samples were loaded on TLC Silica gel 60 F$_{254}$ plate. Glucose + Sucrose mix was used as reference. Laminarin, laminarihexaose and their hydrolysis products were visualized by spraying the TLC plate with detection solution. Arrowheads indicate released of hydrolyzed products. Similar results were observed in three independent biological experiments. **f** The signal intensity of bands representing glucose was quantified by using ChemiDoc Bio-Rad imaging machine. Data are presented as mean value ± SD of three independent biological experiments and asterisks above bars indicate significant differences ($p < 0.05$, two-tailed student's t-test). Calculated $p$ values were presented in Source Data.

Next, we performed a reactive oxygen species (ROS)-burst assay using cellotetraose as an elicitor to test whether Erc1 might interfere with cellotetraose-mediated accumulation of reactive oxygen species in barley leaf discs. Cellotetraose or chitosan (as a positive control) was pre-incubated with or without recombinant Erc1 and tested for the ability to induce a ROS burst on barley leaf discs (Supplementary

Fig. 5c). Chitosan alone induced the typical ROS burst with a peak at 10 min after incubation (Supplementary Fig. 5c). Similar to chitosan, cellotetraose alone also triggered the production of ROS, although the peak of ROS production was detected after 20–30 min incubation (Supplementary Fig. 5c). Erc1 alone also induced a ROS burst on barley leaf discs, but the addition of either chitosan or cellotetraose samples

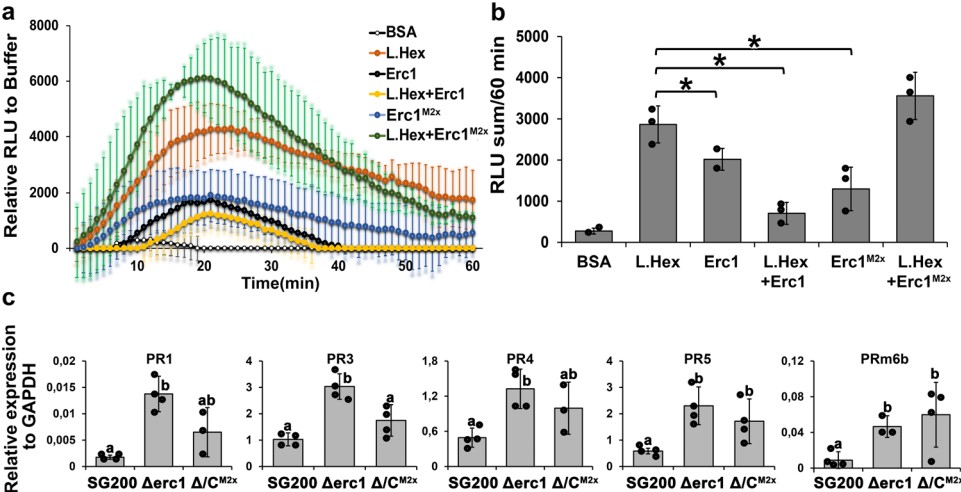

**Fig. 5 | Smut Erc1 prevents induction of host defenses. a** A ROS-burst assay was performed with barley leaf disks following incubation with laminarihexaose, Erc1, Erc1^M2x, and a mix of laminarihexaose and Erc1 recombinant proteins. BSA was used as a negative control for background signal. Relative luminescence units (RLU) indicate ROS burst activity of treated barley leaf discs. The RLU are normalized with the buffer control. Each curve shows the mean of at least nine technical leaf discs with three independent biological experiments. Data are presented as mean value ± SD. **b** Statistical analysis was performed with sum of RLU values of each sample. Data are presented as mean value ± SD. Asterisks above bars indicate significant differences ($p < 0.05$, two-tailed student's $t$ test). **c** RT-qPCR for *pathogenesis related (PR)* gene expression on SG200 and SG200Δerc1 mutant strains infected maize leaves at 4 days post infection (dpi). The expression levels of maize *PR* genes, including *PR1*, *PR3*, *PR4*, *PR5*, and *PRm6b* were calculated relative to the *GAPDH* gene of maize. Data are presented as mean value ± SD. Asterisks above bars indicate significant differences ($p < 0.05$, pairwise Kruskal-Wallis H-test). Calculated *p* values are shown in the Source Data.

to the Erc1 protein resulted in twofold and fivefold increase in ROS burst levels, respectively (Supplementary Fig. 5c).

It is well documented that β−1,3-glucans, such as laminarin and laminarihexaose, can be recognized by plant cells and thereby induce defense responses including a ROS burst[22,23]. To test whether Erc1 interfered with laminarihexaose-induced plant defenses, we performed laminarihexaose-induced ROS burst assays on barley leaf discs (Fig. 5a, b). Laminarihexaose alone induced a stable ROS burst in the barley leaf discs but this burst was significantly reduced by addition of the Erc1 protein (Fig. 5a, b). In contrast, we did not observe a significant difference in the laminarihexaose-induced ROS burst when laminarihexaose was incubated with the mutant protein Erc1^M2x. This indicates that i) the enzymatic activity of Erc1 is required for suppression of the laminarihexaose-induced ROS burst and ii) that the residual activity of Erc1^M2x in the in vitro enzyme assay was not sufficient to block the laminarihexaose-mediated ROS burst (Fig. 5a, b). In addition, we observed that both Erc1 and Erc1^M2x recombinant proteins alone induced a weak ROS burst in barley leaf discs (Fig. 5a, b). While Erc1^M2x appeared to have an additive effect on the laminarihexaose-induced ROS burst, the Erc1 plus laminarihexaose reaction mixture resulted in a residual ROS accumulation similar to that of Erc1 alone, which suggests that the glucan-induced burst was completely abolished (Fig. 5a, b). Complementary to the ROS burst assay in leaf discs, we performed RT-qPCR analysis on *PR*-gene expression in maize leaves 4 days after infection with *U. maydis* SG200, SG200Δerc1, and SG200Δerc1/Erc1^M2x strains (Fig. 5c). The RT-qPCR data revealed a significant increase in the expression level of *PR1*, *PR3*, *PR4*, *PR5*, and *PRm6b* genes in maize leaves infected with *U. maydis* SG200Δerc1 and SG200Δerc1/Erc1^M2x strains compared to SG200-infected leaves (Fig. 5c), indicating that Erc1 is required for prevention/suppression of host immune responses during host colonization. These data suggest that an active Erc1 effector is required to hydrolyze fungal wall-derived β−1,3-glucans, so that these DAMP molecules will not be perceived by the host to induce ROS burst and expression of *PR* genes.

## Discussion

The corn smut fungus *U. maydis* infects all aerial organs of maize plants, including tassel, ears and leaves[7]. It has been shown that *U.*

*maydis* deploys diverse sets of effectors to colonize different maize organs and around 45% of secreted proteins in *U. maydis* behave in an organ-specific manner[8,9]. However, there is little information about what determines the organ-specific function of *U. maydis* effectors. *U. maydis* Erc1 has previously been identified as a leaf-specific virulence factor[8]. In this study, we characterize its function and show that Erc1 is a conserved core effector of smut fungi whose β−1,3 glucanase activity is required for its cell type-specific virulence function.

Erc1 consists of an N-terminal signal peptide for extracellular secretion, an N-terminal CBM and a GH51 domain. Both confocal microscopy and immunogold labeling TEM micrographs confirmed the secretion of Erc1 into the biotrophic interface, which is in line with the presence of a signal peptide. Although TEM micrographs did not show any specific Erc1 accumulation in fungal or plant cell walls, carbohydrate-binding assays revealed that the protein binds to the plant cell wall polysaccharides cellulose and mix-linkage glucan (i.e., lichenan). This does not only confirm the predicted CBM domain of Erc1, but also suggests that it may target maize plant cell wall components rather than the fungal ones. In addition, the importance of the CBM domain for the virulence function of Erc1 further supports the hypothesis that plant cell wall components are the biologically relevant virulence targets of Erc1.

Homology search showed that *U. maydis* has two Erc1 homologs, Afg2 (GH51 family) and Afg3 (GH62 family). All three genes are predicted to encode α-L-arabinofuranosidases and are expressed during host colonization but not in axenic culture[11]. Very high and continuous expression patterns of Erc1 in both *U. maydis*-maize and *U. hordei*-barley pathosystems indicate that Erc1 is required during the whole colonization process of these smut fungi[11,24]. Indeed, deletion of *erc1* in *U. maydis* and *U. hordei* resulted in a significant reduction in fungal virulence, demonstrating that Erc1 is a virulence factor of both smut fungi. Unlike Erc1, neither Afg2 nor Afg3 are required for *U. maydis* virulence[19]. However, the previous study by Lanver et al. also reported that a single deletion of the *Erc1* (named as "*afg1*") gene had no effect on virulence in maize, while deletion of all three predicted *U. maydis* α-L-arabinofuranosidases decreased virulence[19]. We can only speculate that the virulence function of Erc1 had been overlooked by Lanver and colleagues. Similar to our findings, also a recent study by Marin-

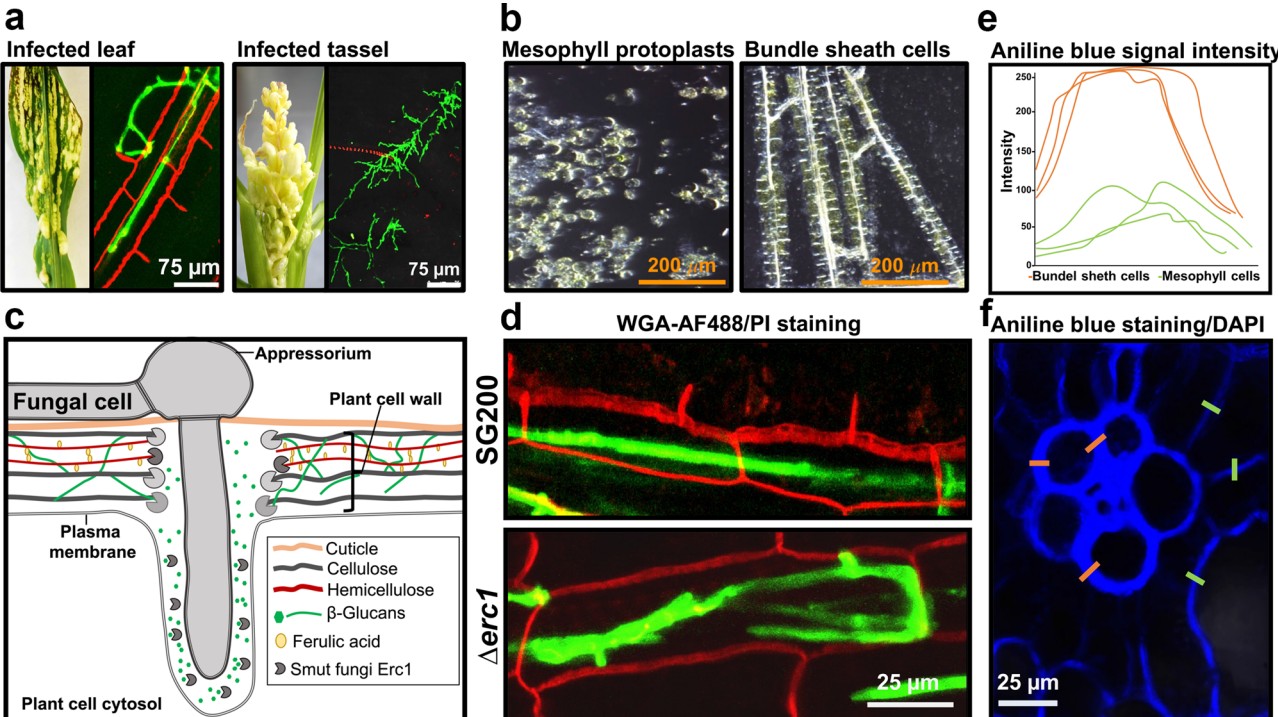

**Fig. 6 | Cell type-specific function of Erc1. a** Microscopic observation of trapped *Ustilago maydis* infected maize leaf and tassel tissues via WGA-AF488/Propidium iodide staining. WGA-AF488 (green color indicates fungal cell wall): excitation at 488 nm and detection at 500-540 nm. PI (red color indicates bundle sheath cells): excitation at 561 nm and detection at 580−630 nm. Unlike leaf tissue, no bundle sheaths cells are detectable in tassel tissue. **b** Protoplastation of maize leaf cells. Pictures were taken after treatment with plant cell wall degrading enzymes. While the used enzyme mix has the ability to convert mesophyll cells into protoplasts, bundle sheath cells remain intact. **c** Schematic depiction of the biotrophic interface of host-smut interaction. Fungus-derived proteins are depicted in gray. During host colonization, 1,3-β-glucan molecules are accumulated at the biotrophic interface and Erc1 may hydrolyze these 1,3-β-glucans to prevent the accumulation and subsequent recognition of DAMP molecules by the host plant. **d** While wild-type smut fungi have the ability to move from cell-to-cell in bundle sheath cells, Δerc1 mutants do not have the ability to fully suppress host immunity leading to the described cell-arrest phenotype (Modified from Fig. 1c). **e, f** Aniline blue staining with maize leaf cross-section for detection of callose. Intensity of aniline blue signal is depicted via graph (**e**). Orange bars indicate bundle sheath cell wall, green bars indicate mesophyll cell wall. Similar results were observed in two independent biological experiments.

Menguiano and colleagues reported a significant contribution of Erc1 in virulence of *U. maydis* CL13 strain[25].

In a previous comparative genome and secretome analysis performed by Schuster et al., Erc1 (Afg1) has been classified as a core effector being highly conserved in smut fungi[11,26]. Our phylogenetic tree analysis shows that, although both Afg2 and Afg3 are present in a wide range of bacterial and fungal species, Erc1 is exclusively present in the Ustilaginomycotina, i.e., the smut fungi. The biotrophic rust fungi, which also belong to the Basidiomycetes and share plant hosts with the smuts, have homologs of Afg2, but they do not possess Erc1 homologs.

While a commercially available *U. maydis* Afg3 can hydrolyze wheat arabino-xylan, suggesting an actual α-L-arabinofuranosidase activity of this enzyme, we could not confirm the predicted α-L-arabinofuranosidase function of Erc1, neither with α-L-arabinofuranosidase specific ABPP probes nor with α-L-arabinofuranosidase specific 4NP-arabinofuranoside substrate. Moreover, complementation of the SG200Δerc1 mutant with *Afg2* and *Afg3* under the control of the *Erc1* promoter could not rescue the decreased virulence phenotype of the *erc1* deletion mutant, indicating different enzymatic activity/virulence functions of these three proteins. Taken together the phylogenetic restriction to the Ustilaginomycotina and the distinct enzymatic activity, Erc1 appears to be a conserved core effector that specifically evolved in smut-host interactions where it functions as a virulence factor.

Smut fungi penetrate and colonize host vascular bundles by moving from cell-to-cell[11,27]. Our microscopy showed that Δerc1 mutants fail in cell-to-cell extension specifically in host bundle sheath cells, demonstrating a cell-type specific function of Erc1. Previous studies reported developmental and structural differences between different plant cell types, including cell wall thickness and composition[28]. Both wheat germ agglutinin (WGA)-AF488/PI staining and protoplasting assays, which are regularly performed with maize leaves in our laboratory, demonstrated that there are some differences in the cell wall composition of bundle sheath and mesophyll cells (Fig. 6a, b). While in the WGA-AF488/PI staining assay only bundle sheath cells could be stained with propidium iodide (PI) (Fig. 6a, left picture, red signal), the cell wall-degrading enzyme mixture that is used in the protoplasting assay has an effect only on mesophyll but not bundle sheath cells (Fig. 6b). We hypothesize that differences in cell wall composition or thickness might be the cause for the cell-type specific function of Erc1. In support of this, aniline blue staining carried out to determine callose composition of different cell types in maize leaf cross section also revealed presence of stronger aniline blue signal in the wall of bundle sheath cells compared to mesophyll (Fig. 6e, f). Furthermore, WGA-AF488/PI staining of tassel tissue following infection with *U. maydis* showed no PI staining, reflecting the absence of bundle sheath cells and similar cell wall composition with mesophyll cells (Fig. 6a, red signal). In this regard, we suggest that the organ-specific function of Erc1 simply results from the absence of bundle sheath cells in maize tassel tissue.

Fungal-derived extracellular members of the GH family and glucan-binding proteins have often been shown to play important roles in plant host penetration. While the repertoire of PCWDEs in necrotrophs is relatively high in number and composition, thus enabling direct maceration and killing of host cells for nutrient acquisition, biotrophs fine-tune their limited number of PCWDEs to

detoxify antimicrobial compounds, to acquire nutrients or to sequester MAMP molecules to prevent induction of host immunity[1]. For example, some members of the GH3 and GH10 family are involved in detoxification of host-specific avenacin and α-tomatine, antimicrobial secondary metabolites from oat and tomato, respectively[13,29,30]. Some GHs have the ability to sequester MAMP molecules that are released from the fungal cell wall during host infection. A GH18 chitinase from *Magnaporthe oryzae* (MoChia1), as well as LysM lectins from a wide range of fungal pathogens bind and sequester chitin fragments released from the fungal cell wall, thereby preventing chitin-triggered immunity[14,31–33].

We found that Erc1 hydrolyzes laminarin, which consists of 20–25 units of 1,3-β-glucan with 1,6-β-glucan linkages, as well as laminarihexaose, which consists of only 1,3-β-glucans. The sole release of glucose monomers from laminarin and the absence of other intermediate-size sugar molecules suggest an exo-β-glucanase activity of Erc1. Site-directed mutagenesis of two predicted active sites of Erc1 in *U. maydis* resulted in a significant reduction in virulence compared to the SG200, supporting the relevance of these two active sites in the virulence function of Erc1. Furthermore, TLC with Erc1[M1x] and Erc1[M2x] recombinant proteins showed that the enzymatic activity of both single- and double-active site mutant was significantly lower compared to the wild-type protein. However, the remaining residual enzymatic activity for both mutant proteins suggests a yet undiscovered active site of Erc1.

1,3-β-glucans are well-described elicitors, inducing basal defense responses such as callose (another 1,3-β-glucan) deposition at infection sites[22,23,34]. Recently, we performed immune gold labeling of 1,3-β-glucans on *U. hordei* infected barley leaf sections; subsequent TEM analysis revealed a strong accumulation of 1,3-β-glucans at the biotrophic interface[35]. Thus, one could hypothesize that Erc1 hydrolyzes 1,3-β-glucans to prevent the accumulation and subsequent recognition of these DAMP molecules at the biotrophic interface (Fig. 6c). This would also be consistent with our observation that Erc1 significantly reduces the laminarihexaose-induced ROS burst and is involved in suppression of *PR* gene expression and this activity depends on its enzymatic activity. Thus, we suggest that Erc1 hydrolyzes 1,3-β-glucans accumulated at the biotrophic interface and in the absence of Erc1 the accumulated 1,3-β-glucans induce defense responses. Additionally, different immune responsiveness of bundle sheet cells might result in the observed cell arrest phenotype of Δerc1 mutants (Fig. 6d). Another explanation for observed cell-arrest phenotype could be due to the composition of bundle sheath cell wall. Bundle sheath cells have more robust walls compared to mesophyll cells, and this could form a stronger physical barrier against any intruder (Fig. 6e, f). Maize leaf cross sections showed higher accumulation of aniline blue staining signals at the wall of bundle sheath cells compared to mesophyll cells, indicating an increased content of 1,3-β-glucans (Fig. 6e, f). This observation also supports the importance of β-glucanase activity of Erc1 in cell-to-cell extension. Although in our study, we found only laminarin as a substrate of Erc1 substrate, one cannot exclude the possibility of additional, yet undetermined targets of Erc1 in the host cell wall which might contribute to restricting fungal cell-to-cell extension in bundle sheath cells.

Although fungal-derived members of the GH family play important roles in fungal virulence, in some cases these proteins or their released products can be recognized as MAMPs. For example, a GH45 cellulase (EG1) from *Rhizoctonia solani*, a GH12 xyloglucanase (BcXyg1) of *Botrytis cinerea* and a GH11 xylanase (EIX) of *Trichoderma viride* are recognized as MAMPs, triggering plant cell death and other immune responses[36–39]. Oligosaccharides released by *Cladosporium fulvum* CfGH17-1 (1,3-β-glucanase) from tomato cell walls trigger cell death upon recognition as DAMPs[40]. Besides the suppression of laminarihexaose-induced ROS burst, Erc1 itself might be recognized by the plant independently of its enzymatic activity. Thus, Erc1 has an additive effect on the ROS-burst-inducing activity of chitosan and cellotetraose elicitors. This additive effect on the induction of ROS-burst implies that Erc1 and carbohydrate elicitors are recognized by different pathways. It also suggests that other effectors are involved in the suppression of Erc1-mediated defense responses. Thus, several questions remain to be answered in upcoming studies: What are the determinants of the cell-type specificity of Erc1? What is the receptor that recognizes Erc1 to induce ROS burst, but also what is the evolutionary origin of Erc1 proteins and how are they involved in the pathogenic lifestyle of smut fungi, i.e., their ability to colonize vegetative plant organs?

In summary, we functionally characterized a cell-type specific Erc1 effector from smut fungi that exhibits 1,3-β-glucanase activity and thus it is also involved in the suppression of 1,3-β-glucan-mediated ROS burst and *PR* gene expression in the host plant. Erc1 is functionally conserved in all plant pathogenic smuts and is required for cell-to-cell extension in bundle sheath cells.

## Methods

### Growth conditions for fungal and bacterial cultures

The *Escherichia coli* DH5α strain was grown in dYT-medium (1.6% w v⁻¹ peptone, 1% w v⁻¹ yeast extract and 0.5% w v⁻¹ NaCl) with appropriate antibiotics at 37 °C with 200 rpm shaking (for liquid cultures). *Ustilago maydis* SG200 and *U. hordei* DS199 solopathogenic strains were incubated in YEPS_light (0.4% w v⁻¹ yeast extract, 0.4% w v⁻¹ peptone, and 2% w v⁻¹ sucrose) liquid medium at 28 °C and 20 °C with 200 rpm shaking, respectively[1]. Growth of *U. maydis* and *U. hordei* cultures on plates was carried out on potato dextrose agar with appropriate antibiotics (concentrations: 200 μg ml⁻¹ hygromycin or 2 μg ml⁻¹ carboxin). The *Pichia pastoris* KM71H-OCH strain was used for recombinant protein expression. YPD medium supplemented with 100 μg ml⁻¹ zeocin was used for the initial growth of *P. pastoris* strains at 28 °C and 200 rpm shaking (for liquid cultures). *Zea mays* L. Early Golden Bantam (EGB) maize (Olds Seeds, Madison, WI, USA) and *Hordeum vulgare* Golden Promise (GP) barley cultivars were grown for infection assays.

### Nucleic acids methods

Plasmid DNA from bacterial cells was isolated using the QIAprep Spin Miniprep Kit (Qiagen; Hilden, Germany) according to the manufacturer's information. The genomic DNA isolation from *U. maydis* and *U. hordei* was performed according to the protocol described by Schultz et al.[41]. The polymerase chain reaction (PCR) utilizing Phusion© polymerase enzyme (Thermo Fisher Scientific; Darmstadt, Germany) was used to amplify specific DNA fragments by using gene specific primer pairs depicted in Supplementary Data 1.

Total RNA isolation was performed with crushed infected leaf material (at 4 dpi) using the TRIzol® extraction method (Invitrogen; Karlsruhe, Germany) according to the manufacturer's instructions. Subsequently, the Turbo DNA-Free™ Kit (Ambion/Applied Biosystems; Darmstadt, Germany) was used to remove any genomic DNA contamination. cDNA synthesis was performed with 1 μg of total isolated RNA by using the First strand cDNA synthesis Kit (Thermo Fisher Scientific; Darmstadt, Germany). RT-qPCR analysis for *PR* gene expression was performed by using SYBR® Green Supermix (BioRad; Munich, Germany). The reaction was performed in a Bio-Rad iCycler system using the following program: 2 min at 95 °C followed by 45 cycles of 30 s at 95 °C, 30 s at 61 °C and 30 s 72 °C. The expression levels of maize *PR* genes were calculated relative to the *GAPDH* gene of maize (*NM001111943*). Results of at least three biological RT-qPCR replicates were analyzed using the $2^{-\Delta\Delta Ct}$ method[42]. The primers used for RT-qPCR are listed in Supplementary Data 1.

### Construction of plasmids

All plasmid constructions were performed by using standard molecular biology methods according to molecular cloning laboratory

manual of ref. [43]. gDNA from *U. maydis* and *U. hordei* were used for the amplification of deletion and complementation constructs via PCR, whereas cDNA was used to amplify expression constructs. After PCR amplification of the desired genomic region with appropriate primers (Supplementary Data 1), the PCR fragments were digested with appropriate restriction enzymes and ligation reaction was performed with the T4-DNA ligase (New England Biolabs; Frankfurt a.M., Germany). All vector constructs, primer pairs, and restriction sites that were used for the cloning procedures are indicated in Supplementary Data 1. The nucleotide sequences of all constructs were confirmed via sequencing at the Eurofins sequencing facility (Germany).

Chemically competent *Escherichia coli* DH5α cells were transformed via heat shock assay according to standard molecular biology methods[43]. *P. pastoris* was transformed as described in the yeast protocol handbook (Clontech, Mountain View, USA). Gene replacement and transformation assays for *U. maydis* and *U. hordei* protoplasts were conducted as described in ref. [44]. All generated deletion and complementation mutant strains were confirmed via southern blot analysis for single integration events in the desired loci (Supplementary Fig. 6).

## CRISPR/Cas9 gene editing system
The CRISPR/Cas9-HF (high fidelity) gene editing system was used to knockout the *Erc1* gene in the *U. hordei* as described in ref. [35]. To express sgRNA for the targeted gene, the *Ustilago maydis pU6* promotor was replaced with the *U. hordei pU6* promotor. The sgRNAs for knocking out the *U. hordei Erc1* gene was designed by E-CRISPR (http://www.e-crisp.org/ECRISP/aboutpage.html) (Supplementary Data 1)[45]. Plasmid construction for CRISPR/Cas9 was performed as described by ref. [46]. The CRISPR/Cas9-HF vector was linearized with the restriction enzyme Acc65I, and subsequently assembled with spacer oligo and scaffold RNA fragment with 3′ downstream 20 bp overlap to the plasmid by using Gibson Assembly[47].

## *Ustilago maydis* virulence assay
The *U. maydis* virulence assays on the EGB maize cultivar were performed as described in ref. [48]. Disease symptoms for *U. maydis* were scored at 12 dpi using the disease rating scheme developed previously[27]. For statistical analysis for *U. maydis* virulence assays, the disease index was calculated as follows: The number of plants sorted into categories "chlorosis", "small tumor", "normal tumor", "big tumor" and "heavy tumor" were multiplied by 1, 3, 6, 9, and 12, respectively. All calculated numbers for each strain were summed and then divided by the total number of infected plants. The *U. hordei* virulence assays on the GP barley cultivar were performed as described in ref. [24]. Infected barley leaves were collected at 8 dpi for gDNA isolation and followed by qPCR to quantify the fungal biomass in mutant and DS199 strains. All virulence assays were performed in three independent biological replicates. A student *t* test was performed to calculate significant differences in disease indices between mutant and solopathogenic strains.

## Heterologous protein production in *Pichia pastoris*
The *Pichia pastoris* KM71H-OCH protein expression system was used to produce N-terminally His and C-terminally Myc-His tagged *U. maydis* Erc1, Erc1$^{M1x}$, and Erc1$^{M2x}$ recombinant proteins. All genes for each respective protein were cloned into the pGAPZαA vector (Invitrogen; Carlsbad, USA) under the control of a constitutive promotor with an α-factor signal peptide for secretion. Protein expression was performed by growing *Pichia* in 1 L buffered (100 mM sodium phosphate buffer, pH 6.0) YPD medium at 28 °C for 48 hours with 200 rpm shaking (pGAPZαA, B, & C *Pichia pastoris* Expression Vectors, Invitrogen; Carlsbad, USA). Recombinant protein purification was performed with a Ni-NTA-matrix (Ni-Sepharose™ 6 Fast-Flow, GE-Healthcare; Freiburg, Germany). After protein purification, each protein sample was applied to the NAP-25 column to exchange buffer with 20 mM potassium

phosphate buffer pH 6.0. The proteins were stored at −20 °C for further experiments. Western blot analysis was performed by using anti-His (H1029, Sigma-Aldrich, Steinheim) and anti-Myc (M4439, Sigma-Aldrich, Steinheim) antibodies with 1:5000 and 1:3000 dilution, respectively[20]. As a secondary antibody, anti-mouse IgG HRP (#7076, Cell Signaling Technology) was used in 1:3000 dilution[20].

## Enzyme activity assay and activity-based protein profiling
To determine the enzymatic activity of the purified Erc1 protein, 4-nitrophenyl α-L-arabinofuranoside (4NPA) (Sigma-Aldrich, Steinheim) was diluted in 0.1 M sodium acetate buffer (pH 4) to concentrations of 1 mM and 5 mM. The 4NPA substrate was incubated with 1.34 and 9 µM Erc1 and heat-inactivated recombinant protein at 40 °C. After 10 min incubation, the reaction was stopped by adding 150 µl 2% trisodium phosphate buffer (pH: 12). The absorption was measured at 400 nm. The commercially available AFASE from *Aspergillus niger* (Megazyme, Ireland) was used as a positive control.

For ABPP assay, 15 µl (80 µg ml⁻¹ stock) of Erc1, Erc1$^{M2x}$, and AFASE recombinant proteins were incubated in 300 mM sodium acetate buffer pH:6.0 for 30 min with and without 50 µM DL69 α-L-arabinofuranosidase specific inhibitor[7]. Subsequently, 5 µM α-L-arabinofuranosidase specific ME868 probe was added to the reaction mixture and incubated for 2 h at room temperature at dark. The reaction was stopped with 6X SDS-loading dye and the probe was detected by scanning the in-gel fluorescence with Cy5 filter (Ex. 650 nm, Em. 670 nm). Protein of loaded samples was visualized via Sypro Ruby (Ex. 450 nm, Em. 610 nm).

## Thin-layer chromatography (TLC) assay
To detect any carbohydrate hydrolyzing activity of Erc1 a TLC assay was performed with different polysaccharides in the presence of recombinant Erc1 protein as described in ref. [40]. Erc1 (50 µl, 500 µg ml⁻¹) recombinant protein was incubated with 50 µl of different carbohydrate solutions (5 mg ml⁻¹) including β-glucan (from barley, Megazyme, lot: 90803b), laminarin (from *Laminaria digitate*, Sigma, L9634), lichenan (from moss, Megazyme, lot: 80402), xylan (from beechwood, Megazyme, lot: 171002) and arabino-Xylan (from wheat, Megazyme, lot: 120601b) overnight at 42 °C. For another TLC assay, 25 µL Erc1 (500 µg ml⁻¹), 25 µl Erc1$^{M1x}$ (500 µg ml⁻¹), and 36 µL Erc1$^{M2x}$ (350 µg ml⁻¹) recombinant protein were incubated with 5 µl of laminarin (from *Laminaria digitate*, Sigma, L9634) and laminarihexaose (Megazyme, lot: 190606) solutions (5 mg ml⁻¹ per each) in 70 µl water overnight at 42 °C. After spinning down the insoluble polysaccharides, 20 µl of each digest was loaded on a TLC Silica gel 60 F$_{254}$ plate (20 × 20 cm) (Merck, HX85205954). Two µL of 3 mg m⁻¹ Glucose + Sucrose mixture was used as reference. Untreated polysaccharide sample and recombinant protein alone samples were used as negative controls. n-propanol:ethanol:water (7:2:1) (v:v:v) was used as running solvent for the TLC assay. Carbohydrates were visualized by spraying a staining solution (45 mg naphthol in 4.8 ml sulfuric acid, 37.2 ml ethanol, 3 ml water) onto the dried TLC plate and subsequent 5–10 min incubation of sprayed TLC plate at 100 °C.

## WGA-AF488/propidium iodide and aniline blue staining
Cell staining with the wheat germ agglutinin (WGA)-AF488 (Molecular Probes, Karlsruhe, Germany) and propidium iodide (Sigma-Aldrich) was performed according to ref. [24]. WGA-AF488 stains fungal cell walls (green), while the propidium iodide stains plant cell walls (red). Briefly, infected leaf material was bleached in pure ethanol and subsequently boiled for 1–2 h in 10% KOH at 85 °C. The pH of the leaf samples was neutralized using 1xPBS buffer (pH: 7.4) with several washing steps. The WGA-AF488/PI staining solution (1 µg ml⁻¹ propidium iodide, 10 µg ml⁻¹ WGA-AF 488; 0.02% Tween 20 in PBS pH 7.4) was vacuum infiltrated into leaf samples three times with a desiccator for 5 min at 250 mbar. WGA-AF488: excitation at 488 nm;

detection at 500–540 nm. PI: excitation at 561 nm; detection at 580–630 nm.

For aniline blue staining, collected maize leaves were first fixed and distained in acetic acid: ethanol (v/v; 1:3) mixture until the samples were transparent. Subsequently, 150 mM $K_2HPO_4$ was vacuum infiltrated in the leaves three times for 10 min. After removing the buffer, samples were covered with 0.05% aniline blue solution (w/v in 150 mM $K_2HPO_4$) and the stain was vacuum infiltrated in the leaves three times for 10 min. Aniline blue infiltrated leaves were incubated at room temperature at dark for 1 hour. After cross-sectioning of the leaf samples, confocal microscopy was carried out with DAPI channel (Emission 490–520 nm).

### Localization of Erc1-mCherry with confocal microscopy

To visualize the secretion of the Erc1-mCherry fusion protein in *U. maydis* during maize colonization, the *U. maydis* SG200 strains expressing Erc1-mCherry, Pit2-mCherry, or cytosolic mCherry were inoculated on maize seedlings. Subsequently, the localization of Erc1-mCherry in infected maize leaves at 4 dpi was analyzed with a Leica confocal microscopy SP8. For detection of mCherry fluorescence of hyphae in maize tissues, an excitation at 561 nm and detection at 580–630 nm were used. Plasmolysis was performed by dropping a 1 M NaCl solution on the leaf sample.

### ROS burst assay

For ROS burst assay, 12 leaf disks, which were dissected from the second leaf of 13 days old Golden Promise barley cultivar, were incubated with 200 µl water in a 96 well plate (Thermo Fisher Scientific-Nunclon 96 Flat Bottom White Polystyrene) overnight in the dark. The next day the water was replaced with a 150 µl reaction mixture containing a luminol solution (100 µM L-012 and 20 µg ml$^{-1}$ HRP). The following reaction mixtures were used in the ROS burst assay for barley leaf disks: (1) Buffer 20 mM $KPO_4$ (pH6) (2) BSA, (3) 250 µM laminarihexaose, (4) 3 µM Erc1, (5) 250 µM laminarihexaose incubated with 3 µM Erc1, (6) 3 µM Erc1$^{M2x}$, (7) 250 µM laminarihexaose incubated with 3 µM Erc1$^{M2x}$. The luminescence from each well was measured with the following settings in a Tecan Infinite 200 Pro plate reader: Kinetic cycle: 60, interval time: 1 min, integration time: 450 ms.

### High-pressure freezing, freeze substitution, and resin embedding for transmission electron microscopy and immunogold labeling

For TEM observations, 2 mm leaf discs from maize leaves infected with *U. maydis* expressing Erc1-HA or GFP-HA under the control of the native Erc1 promoter and signal peptide, were excised from 1 cm below the infected area using a biopsy punch and processed by means of high-pressure freezing (HPF) as described in ref. 49. Freeze substitution (FS) was performed in 0.5% uranyl acetate in acetone (w/v) in a Leica EM AFS2 freeze substitution device (Leica Microsystems GmbH) over 7 days with temperatures ranging from −85 °C (90 h) over −60 °C (24 h) and −30 °C (24 h) to −20 °C (12 h). At the end of the FS run, samples were rinsed in acetone and gradually transferred into ethanol into a Leica EM AFS2 at −20 °C and then carefully removed from their HPF carriers in ice-cold ethanol with the help of a stereomicroscope. Subsequent infiltration in medium-grade LR White resin (Plano GmbH) was gradually performed over 7 days in a freezer at −20 °C with the help of a laboratory rocker; LR White polymerization with UV light in the Leica EM AFS2 was achieved for 24 h at −20 °C and 24 h at 0 °C.

### Sectioning, immunogold labeling, and transmission electron microscopy

Ultrathin (70–90 nm) sections were collected on nickel slot grids as described by ref. 50. For immunogold labeling, sections were blocked for 30 min in a 1:30 dilution of goat normal serum in TRIS buffer (20 mM TRIS, 15 mM NaN3, 225 mM NaCl, pH 6.9) supplemented with 1% (w/v) BSA (Sigma-Aldrich A3294) and 1% (w/v) fish gelatin (FG; Sigma-Aldrich G7765). After three washes for 10 min in TRIS-BSA-FG, sections were incubated in a 1:500 dilution of monoclonal mouse anti-HA antibody (Sigma Aldrich H-9658) for 1 h at room temperature (slow orbital shaking). After washing in TRIS-BSA-FG (4× 10 min), sections were incubated in a 1:20 dilution of secondary goat anti-mouse antibody conjugated to 10 nm colloidal gold particles (British Biocell International, Cardiff, UK) for 1 h (slow orbital shaking). Finally, sections were rinsed in TRIS-BSA-FG (4 × 5 min) followed by a stream of sterile-filtered, distilled water for 3 min. After drying at room temperature, sections were examined without further staining in a Hitachi H-7650 TEM (Hitachi High-Technologies Europe GmbH, Krefeld, Germany) operating at 100 kV fitted with an AMT XR41-M digital camera (Advanced Microscopy Techniques, Danvers, USA).

### Plant cell wall analysis

Plant cell walls were isolated from mock, SG200, and SG200Δerc1 mutant treated EGB maize leaves at 9 dpi. Inoculated plant tissues were lyophilized and homogenized using a MM400 mixer mill (Retsch Technology). Preparation of a destarched alcohol insoluble residue (cell wall preparation) was performed as described in ref. 51. The monosaccharide composition of that residue was analyzed via high-performance anion-exchange chromatography coupled with pulsed electrochemical detection, as previously described[52] using a 940 Professional IC Vario ONE/ChS/PP/LPG instrument (Metrohm) equipped with CarboPac PA20 guard and analytical columns. The monosaccharides were quantified based on known concentrations of standards.

### Bioinformatics methods

The SignalP 5 software program was used to predict an N-terminal secretion signal peptide within a protein sequence (http://www.cbs.dtu.dk/services/SignalP/)[53]. For protein domain prediction the website Pfam (http://pfam.xfam.org/) was used. A minimum evolution tree was constructed by using an alignment of the full-length amino acid sequence of Erc1 homologs which were obtained from the NCBI database for different microorganisms. The minimum evolution tree was constructed by using the software Mega7 (http://www.megasoftware.net/mega.php) with a minimum evolution algorithm performing 1000 bootstraps.

### Reporting summary

Further information on research design is available in the Nature Research Reporting Summary linked to this article.

## Data availability

All data that support the findings of this study which are not directly available within the paper (and its supplementary information files) will be available from the corresponding authors (GD, BÖ) upon reasonable request. Source data are provided with this paper.

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

## Acknowledgements

The authors would like to thank Katharina Lufen, Institute for Plant Cell Biology and Biotechnology, Heinrich Heine University, Düsseldorf for excellent technical assistance with respect to the monosaccharide compositional analysis. GD and MP acknowledge support from the Cluster of Excellence on Plant Sciences (CEPLAS) funded by the Deutsche Forschungsgemeinschaft (DFG, German Research Foundation) under Germany's Excellence Strategy – EXC 2048/1 – Project ID: 390686111. The authors would also like to thank Ila Rouhara from Central Microscopy, Max-Planck-Institute for Plant Breeding Research for excellent technical assistance with respect to TEM. The authors thank Prof. Dr. Hermen Overkleeft from Leiden Institute of Chemistry, LIC/Chemical Biology (The Netherlands) for providing ABPP probes and inhibitors. The authors thank Dr Amey Redkar from the Department of Biology at Savitribai Phule Pune University for kindly providing infected tassel pictures. The authors also thank Dr. Libera Lo Presti from the Cluster of Excellence CMFI University of Tübingen for critical reading and constructive comments on the manuscript.

## Author contributions

B.Ö. and G.D. conceived the project. B.Ö., E.J., L.S., N.F., Y.J.L., A.K., and R.W. carried out the transformation, disease assays, protein production, microscopy, ROS and TLC assays; U.N. carried out the immunogold labeling followed by TEM assay; M.P. carried out cell wall analyses; B.Ö. wrote the manuscript with input from all authors.

## Funding

## Competing interests

The authors declare no competing interests.
