## [Peer Review File · Nature Communications]

A conserved enzyme of smut fungi facilitates cell-to-cell extension in the plant bundle sheathReviewer #1 (Remarks to the Author):

In this manuscript, the authors deeply characterized a conserved effector Erc1 (enzyme required for cell-to-cell movement) of smut fungus *Ustilago maydis*. They show Erc1 is a cell type-specific effector which contributes to fungal virulence in maize leaves, but not on the tassel. Erc1 is required for fungal cell-to-cell movement/extension in the plant bundle sheath. And this cell type-specific function of Erc1 is conserved in the barley pathogen *Ustilago hordei*. They uncovered that Erc1 has a 1,3-b-glucanase activity and inhibits b-glucan-induced defense responses in host leaves. At last, they show that the cell type-specific function of Erc1 is likely due to different cell wall composition between mesophyll cells and bundle sheath cells.

This work is original with respect to the novel enzyme activity of Erc1 which determines its organ-specific function on host. Methods are appropriate and data of high quality.

I have only a few comments listed as follows:

1. The authors used "cell-to-cell movement" to describe the fungal hypha growing from one cell to another cell. Is "extension" better for describe this. Bacteria, virus move from one place to another place, but fugal hypha extend.
2. line 96 the subtitle "UMAG_01829 encodes an enzyme which is required...", could be just "UMAG_01829 is required ...". Or "UMAG_01829 encodes a putative enzyme which ...". Because in the section, no strong data to support that UMAG_01829 encodes an enzyme.
3. Is Erc1 also required for fungal penetration on leaves? It is not very sure whether Erc1 enzyme activity contribute fungal virulence simply by suppressing 1,3-b-glucans induced defense responses, or Erc1 also reduces physical barrier of the cell wall of the bundle sheath cell.

Reviewer #2 (Remarks to the Author):

This is a nice study focuses on virulence mechanism of the *Ustilago maydis* effector Erc1. An interesting feature of this effector is that it is required for cell-to-cell movement specifically in leaves but not tassel. The authors conducted a number of interesting studies to show that Erc1 is functionally conserved in *Ustilago hordei*, secreted into apoplast, can bind plant cell wall components, and possesses 1,3- β -glucanase activity. Erc1 exerts virulence function in a manner dependent on the carbohydrate binding domain and catalytic activity. They additionally showed evidence that Erc1 degrades laminarin to prevent immune responses in the host. Overall, the presented work is solid and interesting. However, the readers are left unexplained why the 1,3- β -glucanase activity is specifically required for virulence in bundle sheath cells but not in tassel. Is it because callose deposition is specifically induced in bundle sheath cells that the fungus must degrades it for cell-to-cell movement? Or is it because of different immune responsiveness to 1,3- β -glucan in different cell types?

Specific comments:

1. What do we know about 1,3- β -glucan composition in bundle sheath cells and tassel? There are commercial antibodies could be used for this.
2. More importantly, callose, the main source of 1,3- β -glucan, is commonly induced at the point of fungal penetration. The authors should compare callose deposition in leaves and tassel during infection by wild-type and enzyme-dead mutant strains of *U. maydis*. Do leaves produce stronger callose deposition than tassel during fungal infection? Does Erc1 counter callose deposition?
3. Does laminarin induce defenses differently in leaves and tassels?
4. Fig 5b, why L.Hex+Erc1 treatment had less ROS than Erc1 treatment alone?
5. Fig 5c, increased expression of PR genes was shown for *erc1* mutant strain. Mutant strain complemented with catalytic mutants are needed to support a role of 1,3- β -glucanase in the suppression of PR gene expression.
6. Line 73, "untypical" should be "atypical".
7. Line 201, "unspecific" should be "nonspecific".

Reviewer #3 (Remarks to the Author):

This manuscript by Ökmen et al. describes the leaf specific virulence factor Erc1 and the mechanism. The main findings are: (1) Erc1 is important for cell-to-cell movement in bundle sheath cells in *U. maydis* and *U. hordei* during host colonization; (2) Erc1 does not exhibit α -L-arabinofuranosidase activity but exhibits exo- β -1,3-glucanase activity and is involved in suppression of 1,3- β -glucan-mediated ROS burst in the host plant, which has previously not been studied. The manuscript suffers from several deficiencies that need, in my opinion, to be addressed before I would recommend this manuscript for publication.

First, also the important: the authors need to explain or comment on the reasons for the difference between the virulence test results of Erc1 from previous pathogenicity tests by Lanver et al. (2014) and Marín-Menguiano et al. (2019). Note that the test by Marín-Menguiano et al. (2019) with SG200 strain also showed that a single deletion of the Erc1 (named as "afg1") gene had no effect on virulence in maize, while single deletion of the Erc1 in CL13 resulted in a significant reduction in fungal virulence. However, you used SG200 strain, and the maize growth conditions, inoculation method and disease scoring you described in the manuscript is similar to them.

Next, the authors need to explain why the ROS-burst assay is only carried out on barley. Also, the arrangement and presentation of data to pictures should be more reasonable. There should be no duplication and unnecessary data. E.g. The mode diagram of the right part in Fig. 1a should be in the supplemental data. Figure 1c and figure 6d are the same.

Last, please use correct statistical methods to analyze relevant data, otherwise the significance of the results is questionable. E.g. the statistical data of disease symptoms for comparison was disease rates, which is not suitable for use One-way ANOVA method. Chi square test are needed in (f) in Fig.2. Nonparametric test is more suitable for comparative analysis of expression level.

As mentioned above, the text needs significant revision. Otherwise, it is not recommended to publish.

Furthermore, the author needs to be more careful with the details of the manuscript. A few minor points listed below:

line 69, a leaf-specific factor was not confirmed in this manuscript.

Afu2 and Afu3 need to be Afg2 and Afg3.

Line 321~325 Pay attention to the test details and conclusions of the reference.

Line 321~325 Pay attention to the test details and conclusions of the reference.

Response to Reviewer Comments:

Reviewer #1:

In this manuscript, the authors deeply characterized a conserved effector Erc1 (enzyme required for cell-to-cell movement) of smut fungus *Ustilago maydis*. They show Erc1 is a cell type-specific effector which contributes to fungal virulence in maize leaves, but not on the tassel. Erc1 is required for fungal cell-to-cell movement/extension in the plant bundle sheath. And this cell type-specific function of Erc1 is conserved in the barley pathogen *Ustilago hordei*. They uncovered that Erc1 has a 1,3-b-glucanase activity and inhibits b-glucan-induced defense responses in host leaves. At last, they show that the cell type-specific function of Erc1 is likely due to different cell wall composition between mesophyll cells and bundle sheath cells.

This work is original with respect to the novel enzyme activity of Erc1 which determines its organ-specific function on host. Methods are appropriate and data of high quality.

I have only a few comments listed as follows:

1. The authors used “cell-to-cell movement” to describe the fungal hypha growing from one cell to another cell. Is “extension” better for describe this. Bacteria, virus move from one place to another place, but fungal hypha extend.

R1: We thank the Reviewer for pointing out this terminology mistake. Indeed, by “cell-to-cell movement” we meant “cell-to-cell extension”. We have changed all “cell-to-cell movement” terms to “cell-to-cell extension” in the whole manuscript.

2. line 96 the subtitle “UMAG_01829 encodes an enzyme which is required...”, could be just “UMAG_01829 is required ...”. Or “UMAG_01829 encodes a putative enzyme which ...” Because in the section, no strong data to support that UMAG_01829 encodes an enzyme.

R2: The text has been changed according to the Reviewer comment.

3. Is Erc1 also required for fungal penetration on leaves? It is not very sure whether Erc1 enzyme activity contribute fungal virulence simply by suppressing 1,3-b-glucans induced defense responses, or Erc1 also reduces physical barrier of the cell wall of the bundle sheath cell.

R3: We have observed an impaired cell-to-cell extension only in host bundle sheets cells, as it is shown in the manuscript. With regard to epidermal penetration: in our lab we test the early phenotypic development (filamentation, appressorium formation, epidermal penetration) of fungal mutants as a standard procedure and the deletion of *erc1* did not cause any defects in these stages. However, the question by this reviewer convinced us that this is actually relevant information for the reader, which was not shown in the previous version of the manuscript. We therefore added the data for appressoria formation and penetration of maize epidermis cells in the new **Figures S2C& S2D** and thank the reviewer for this question.

Apart from this - in our study we show that Erc1 has the ability to hydrolyze and thereby suppress/prevent 1,3-b-glucans-mediated host defense responses. However, this does not exclude that Erc1 is not involved in reducing physical cell wall barrier in bundle sheath cells. Although we determined laminarin as substrate of Erc1, one cannot exclude that Erc1 may also target other unknown plant cell wall components and that this could be relevant for fungal cell-to-cell extension. Therefore, we hypothesize that Erc1 could have both of these functions and collectively contributes to fungal cell-to-cell extension and virulence. We have addressed these arguments and revised this part of the discussion (**lines 439ff**) accordingly.

Reviewer #2 (Remarks to the Author):

This is a nice study focuses on virulence mechanism of the *Ustilago maydis* effector Erc1. An interesting feature of this effector is that it is required for cell-to-cell movement specifically in leaves but not tassel. The authors conducted a number of interesting studies to show that Erc1 is functionally conserved in *Ustilago hordei*, secreted into apoplast, can bind plant cell wall components, and possesses 1,3- β -glucanase activity. Erc1 exerts virulence function in a manner dependent on the carbohydrate binding domain and catalytic activity. They additionally showed evidence that Erc1 degrades laminarin to prevent immune responses in the host. Overall, the presented work is solid and interesting.

However, the readers are left unexplained why the 1,3- β -glucanase activity is specifically required for virulence in bundle sheath cells but not in tassel. Is it because callose deposition is specifically induced in bundle sheath cells that the fungus must degrades it for cell-to-cell movement? Or is it because of different immune responsiveness to 1,3- β -glucan in different cell types?

R: Callose deposition in plant cell occurs upon induction of plant immunity and this accumulation is not bundle sheath cell-specific. Both epidermis and mesophyll cells have the ability to accumulate callose at the site of infection. We hypothesized - and showed - that the cell wall composition/thickness of bundle sheath cells is different from mesophyll cells. In this regard, we think there are several explanations for cell type specific function of Erc1: first, it could be that bundle sheath cells have higher amount of callose production (it is known that the cell wall of bundle sheath is much thicker and more robust compared to mesophyll cells) and thereby there is more defense responses in case of Erc1 mutant. Consistently, aniline blue staining of maize leaf cross section showed higher accumulation of signal for presence of callose at the cell wall of bundle sheath cells (**new figure Fig. 6e-f**). Second, like this reviewer suggested, there could be different immune responsiveness to 1,3- β -glucan in different cell types. And last, although in this study we determined only laminarin as a substrate of Erc1, there could be still un-detected plant cell wall components that is targeted by Erc1. In this regard, Erc1 may be involved in reducing physical cell wall barrier in bundle sheath cells to lead in fungal cell-to-cell extension.

We addressed this topic in the updated discussion part (**lines 439ff**).

Specific comments:

1. What do we know about 1,3- β -glucan composition in bundle sheath cells and tassel? There are commercial antibodies could be used for this.

R1: In this study we hypothesized that the cell-type specificity of Erc1 is the main determinant of its organ specificity as well. Consequently, we conclude that the *erc1* mutant does not have an impaired virulence phenotype in tassel, simply because bundle sheath cells are absent in this tissue. Although our propidium iodide staining showed that there are some differences in staining of the wall of bundle sheath, mesophyll, and tassel cells, we could not completely profile the cell wall composition of tassel cells. We actually have carried out immuno-gold labeling for 1,3- β -glucan composition of maize leaf, but from our TEM micrographs we could not observe any significant difference in 1,3- β -glucan composition of different maize leaf cells (data not shown in the manuscript, **Figure R1, below**). However, we additionally have performed aniline blue staining in maize leaf cross section, which resulted in higher accumulation of signal in the cell wall of bundle sheath cells compared to mesophyll cells, in which we did not observe any cell trap phenotype (**new figure Fig. 6e,f**).

In addition, the *U. hordei* Δ *erc1* mutant is also impaired in virulence and shows the cell-trapped phenotype in bundle sheath cells of barley, a plant that does not form a tassel, which further demonstrates the cell-type specificity of the Erc1 effector.

Figure R1. Immuno gold labeling (with 1,3- β -glucan specific antibody) and TEM analysis of *Ustilago maydis* infected maize leaf cross section.

2. More importantly, callose, the main source of 1,3- β -glucan, is commonly induced at the point of fungal penetration. The authors should compare callose deposition in leaves and tassel during infection by wild-type and enzyme-dead mutant strains of *U. maydis*. Do leaves produce stronger callose deposition than tassel during fungal infection? Does Erc1 counter callose deposition?

R2: To address this point, we have tested the accumulation of callose in maize leaves that were infected with *U. maydis* wild type or the *erc1*-mutant, respectively. However, we did not see a difference in callose deposition, which implies that callose-deposition is not affected by presence/absence of Erc1 (**Figure R2, just below**). In this regard, we did not extent the analysis to tassel, as one could not expect any different in this tissue (where Erc1 is not involved in fungal virulence).

Figure R2: Aniline blue staining to detect callose deposition in SG200 and Δ erc1 infected maize leaves. Yellow arrow heads show fungal hyphae.

3. Does laminarin induce defenses differently in leaves and tassels?

R3: We have performed (well, we tried) ROS burst assay with maize leaves as well; however, we have observed an overall quite low levels of ROS-burst response and a very high variability in the assays, which made the data unreliable/statistically not robust for the laminarin- or laminarihexaose assays in maize. Due to these experimental issues in ROS burst assays in maize, we used barley leaves for ROS burst assay, which resulted in very robust, reproducible results. Since Erc1 has the same, cell-type specific function in barley leaves we think that this is not affecting the main message of our work (of course, we have to admit that we cannot provide the comparison to tassel, which not exist in barley)

4. Fig 5b, why L.Hex+Erc1 treatment had less ROS than Erc1 treatment alone?

R4: In the bar graph (Fig 5b), we used average of the all-data points for each treatment and since we did exclude all negative values (luminescence cannot have negative values), the Erc1 Erc1+L.Hex treatment has less data points compared to Erc1 and Erc1^{M2x}. This may lower the values in the Fig 5b. Although in the graphical depiction of the data there is not that difference between Erc1 and Erc1+L.Hex treatment (Fig. 5a).

5. Fig 5c, increased expression of PR genes was shown for erc1 mutant strain. Mutant strain complemented with catalytic mutants are needed to support a role of 1,3- β -glucanase in the suppression of PR gene expression.

R5: We have performed new RT-qPCR including catalytic death mutant of Erc1 and **updated Fig. 5c and text accordingly**. Results showed that catalytic death mutant has a like-mutant or an intermediate phenotype.

6. Line 73, “untypical” should be “atypical”.

R6: “untypical” has been changed with “atypical”.

7. Line 201, “unspecific” should be “nonspecific”.

R7: “unspecific” has been changed with “nonspecific”.

Reviewer #3 (Remarks to the Author):

This manuscript by Ökmen et al. describes the leaf specific virulence factor Erc1 and the mechanism. The main findings are: (1) Erc1 is important for cell-to-cell movement in bundle sheath cells in *U. maydis* and *U. hordei* during host colonization; (2) Erc1 does not exhibit α -L-arabinofuranosidase activity but exhibits exo- β -1,3-glucanase activity and involved in suppression of 1,3- β -glucan-mediated ROS burst in the host plant, which has previously not been studied. The manuscript suffers from several deficiencies that need, in my opinion, to be addressed before I would recommend this manuscript for publication.

1-First, also the important: the authors need to explain or comment on the reasons for the difference between the virulence test results of Erc 1 from previous pathogenicity tests by Lanver et al. (2014) and Marín-Menguiano et al. (2019). Note that the test by Marín-Menguiano et al. (2019) with SG200 strain also showed that a single deletion of the Erc1 (named as “afg1”) gene had no effect on virulence in maize, while single deletion of the Erc1 in CL13 resulted in a significant reduction in fungal virulence. However, you used SG200 strain, and the maize growth conditions, inoculation method and disease scoring you described in the manuscript is similar to them.

R1: Indeed, in their publication Lanver et al. (2014) did not report a virulence defect for Erc1 in SG200 *U. maydis*. The reviewer is also absolutely right, that the Lanver-study used the same fungal strain. Thus, we can only speculate why they have not observed a virulence defect for Erc1. One reason could be technical differences in disease assay and scoring. Moreover, although we have used similar (but not identical) growth conditions, it is evident that, for example, at different season of the year, infection efficiency of *Ustilago maydis* can change. For example, in our lab we avoid to perform *U. maydis* infection assays in July/August, because despite controlled conditions in the greenhouse, infection assays show increased variability in the summer which makes it difficult to obtain robust data for quantitative phenotypes.

As stated above – we can only speculate on the reasons of a different results obtained in previous studies. It should also be mentioned that the virulence function of Erc1 (in a leaf-specific manner) has already been found by our lab previously (see Schilling et al, 2014), as it is mentioned in the introduction of the manuscript (line 28ff).

While we are responsible for the reliability of our own data (and the infection data presented in this study is highly robust and results from multiple, independent biological replicates in two different pathosystems), we cannot judge about all possible reasons for other results being published previously. However, this also should remind us that data on microbial virulence that is obtained under controlled laboratory conditions, will never cover the whole complexity of a natural host-microbe interaction, as it always reflects the specific conditions that were used in an experiment.

2-Next, the authors need to explain why the ROS-burst assay is only carried out on barley.

R2: As outlined above (Reviewer 2, R3): we have performed ROS burst assays with maize as well; however, due to the high variability of data, in our hands this did not result reliable results. Contrary, in our hands the responses to laminarin- or laminarihexaose were very robust and highly reproducible in barley leaves. Since the cell-type specific virulence function of Erc1 is similar in both plants, we think that this is not affecting the main message of our work.

3-Also, the arrangement and presentation of data to pictures should be more reasonable. There should be no duplication and unnecessary data. E.g. The mode diagram of the right part in Fig. 1a should be in the supplemental data. Figure 1c and figure 6d are the same.

R3: We thank the reviewer for pointing this out! Since Fig. 6 was prepared as a model to summarize our hypothesis, we had used same picture from 1c. However, we fully agree with reviewer and therefore have replaced the image in Figure 6d.

With regard to the scoring sheet in Figure 1a: scoring of *U. maydis* is performed based on its tumor formation severity and readers who are not familiar with *U. maydis*-maize pathosystem, could have difficulty to imagine the described phenotype just by seeing bar graph. Therefore, we'd prefer to keep the disease scoring panel in **Fig 1a** to show readers who are not familiar with the system, how *U. maydis* symptoms looks like.

4-Last, please use correct statistical methods to analyze relevant data, otherwise the significance of the results is questionable. E.g. the statistical data of disease symptoms for comparison was disease rates, which is not suitable for use One-way ANOVA method. Chi square test are needed in (f) in Fig.2. Nonparametric test is more suitable for comparative analysis of expression level.

R4: We thank the reviewer for this constructive comment on our statistical analysis. We fully agree and therefore have **re-analyzed the statistical test for Fig. 1b and Fig. 2f** according to Reviewer's suggestion. Furthermore, the **new RT-qPCR results** were also re-analyzed by using Kruskal-Wallis nonparametric test.

All statistical analyses in disease assays were performed with disease index (not with disease rate) that was calculated according to 'Supplementary Information Section *Ustilago maydis* virulence assay' and therefore it is suitable for One-way ANOVA.

5-As mentioned above, the text needs significant revision. Otherwise, it is not recommended to publish. Furthermore, the author needs to be more careful with the details of the manuscript.

R5: We carefully revised the text and made several editorial changes throughout the manuscript.

R6 A few minor points listed below:

line 69, a leaf-specific factor was not confirmed in this manuscript.

We already have shown in Schilling *et al.*, 2014 that Erc1 is a leaf-specific virulence factor. Therefore this introductory statement only repeats what we had shown previously.

6-Afu2 and Afu3 need to be Afg2 and Afg3.

Line 321~325 Pay attention to the test details and conclusions of the reference.

R6: Afu2 & Afu3 were changed with Afg2&3.

Discussion has been modified according to reviewer's comment.

Reviewer #1 (Remarks to the Author):

The authors have addressed all my concerns.

Reviewer #2 (Remarks to the Author):

The authors have done an excellent job in revising the manuscript. They have performed necessary experiments to address the majority of comments. This reviewer is impressed with all the effort made by the authors. The work is an important advance in the field of fungal pathogenesis and suitable for publication in Nature Communications.

Reviewer #3 (Remarks to the Author):

The author has answered the questions and revised the paper as required. I have no other opinions.